# Biosensing Dopamine and L-Epinephrine with Laccase (*Trametes pubescens*) Immobilized on a Gold Modified Electrode

**DOI:** 10.3390/bios12090719

**Published:** 2022-09-03

**Authors:** Mariya Pimpilova, Kalina Kamarska, Nina Dimcheva

**Affiliations:** 1Faculty of Chemistry, University of Plovdiv “Paisii Hilendarski”, 24, Tsar Assen St., 4000 Plovdiv, Bulgaria; 2Laboratory on Biosensors, Centre for Competence PERIMED, University of Plovdiv, 21, Kostaki Peev St., 4000 Plovdiv, Bulgaria; 3Department of Mathematics, Physics and Chemistry, Technical University-Sofia, Branch Plovdiv, 4000 Plovdiv, Bulgaria

**Keywords:** laccase from *Trametes pubescens*, electrodeposited porous gold, covalent binding, dopamine, L-epinephrine, mediated electron transfer (MET)

## Abstract

Engineering electrode surfaces through the electrodeposition of gold may provide a range of advantages in the context of biosensor development, such as greatly enhanced surface area, improved conductivity and versatile functionalization. In this work we report on the development of an electrochemical biosensor for the laccase-catalyzed assay of two catecholamines—dopamine and L-epinephrine. Variety of electrochemical techniques—cyclic voltammetry, differential pulse voltammetry, electrochemical impedance spectroscopy and constant potential amperometry have been used in its characterization. It has been demonstrated that the laccase electrode is capable of sensing dopamine using two distinct techniques—differential pulse voltammetry and constant potential amperometry, the latter being suitable for the assay of L-epinephrine as well. The biosensor response to both catecholamines, examined by constant potential chronoamperometry over the potential range from 0.2 to −0.1 V (vs. Ag|AgCl, sat KCl) showed the highest electrode sensitivity at 0 and −0.1 V. The dependencies of the current density on either catecholamine’s concentration was found to follow the Michaelis—Menten kinetics with apparent constants *K_M_^app^* = 0.116 ± 0.015 mM for dopamine and *K_M_^app^* = 0.245 ± 0.031 mM for L-epinephrine and linear dynamic ranges spanning up to 0.10 mM and 0.20 mM, respectively. Calculated limits of detection for both analytes were found to be within the sub-micromolar concentration range. The biosensor applicability to the assay of dopamine concentration in a pharmaceutical product was demonstrated (with recovery rates between 99% and 106%, *n* = 3).

## 1. Introduction

Laccase is a complex copper-containing metalloproteinase found in trees, fungi and bacteria that exhibits enzymatic activity in the oxidation of aromatic or non-aromatic organics as well as some metallic ions in the presence of molecular oxygen [1]. The enzyme possesses two active sites comprising copper ions coordinated by amino acid residues from the protein shell: T1—paramagnetic “blue” mononuclear copper, responsible for the binding of organic substrate; and T2/T3—tri-nuclear copper cluster that catalyzes the reduction of molecular oxygen to water [2]. Its catalytic activity towards a broad range of substrates predetermines a large variety of industrial applications (e.g., food, textile, biofuel production) as well as soil and water bioremediation [1].

Laccases immobilized on conductive materials have been found to be among the few enzymes capable of exchanging electrons with underlying electrode surface while electrocatalytically reducing oxygen to water via a four-electron mechanism [3]. This point has been explored extensively by numerous researchers in their efforts to develop an efficient cathode for the development of biological fuel cells [4,5,6,7,8,9]. Proper enzyme orientation during its immobilization on the electrode surface guaranteeing electron exchange between the laccase active site and underlying electrodes (direct electron transfer, DET mode) while reducing electrochemically oxygen to water has been often considered a crucial factor for improving reaction efficacy [6,7,8,9].

Another way to improve the efficiency of the oxygen reduction reaction (ORR), even when laccase is immobilized in random orientation, is to add its substrates to mediate the electron transfer with the electrode surface (mediated electron transfer, MET mode). The resulting current rises proportionally to the mediator concentration, because the latter donates one electron to the T1 laccase active site; electricity flows further through internal pathways to the tri-nuclear cluster where the ORR takes place, this way facilitating the electron exchange with the electrode. Mediated oxygen reduction has been actively explored in an electroanalytical aspect as well, since diverse biosensors for phenols [10,11,12,13,14,15], including Bisphenol A [16], catechols [17,18], tartrazine [19] and hydrocinnamic acids [20], have been reported recently. Moreover, the sensitivity of the determination depends not only on the enzyme immobilization strategy, but also on the affinity of laccase to the analyte. In general, the analyte that is oxidized entirely biocatalytically when interfacing the immobilized enzyme, is further reduced electrochemically at the electrode, poised at some reductive potential—usually −0.1 V to −0.5 V (vs. Ag|AgCl) [11].

Dopamine and epinephrine are structurally related catecholamines, playing the role of neurotransmitters (NT) in the human body, the abnormal levels of which in blood plasma or urine often indicate pathological conditions such as tumors or serious neurological disorders [21]. Despite the importance of NTs for clinical diagnosis, the existing quantification methods still need improvements in terms of selectivity [22], sensitivity, flexibility and simplification [23]. In an attempt to overcome the limitations inherent to the routinely used diagnostic kits for the determination of catecholamines, chromatography [24,25], chemiluminescence [26], electrophoresis [27] or electroanalysis with chemically modified electrodes [28,29,30,31] have been proposed.

Electrochemical biosensors offer a range of advantages over these techniques, such as selectivity, sensitivity and low limits of detection as demonstrated by Alvarez and Ferapontova [32] with their RNA-based dopamine aptasensor showing sub-micromolar detection limit. Furthermore, a variety of enzyme-based catecholamine biosensors employing monoamine oxidase [33], cellobiose dehydrogenase [34], PQQ-dependent glucose dehydrogenase [35] and polyphenol oxidases [36] have been reported recently.

Both catecholamines—dopamine and L-epinephrine—are organic substances of pharmaceutical importance since both compounds can be found on the drug market in the form of injection solutions. Therefore, the development of rapid and sensitive methods for their quantification is of immediate practical importance for pharma industry.

In this work we report on the development and optimization of an electrochemical biosensor based on laccase from *Trametes pubescens* immobilized on glassy carbon electrode modified with nanoporous gold. The main benefits from using nanogold-structured electrodes are the boosted sensitivity of the determination due to the enhanced surface area, and improved limits of detection resulting from the improved interfacial conductivity [37]. These two advantages were further explored in designing highly sensitive dopamine biosensors based on laccase immobilized on gold nanoparticles [38,39]. The main disadvantage of these biosensors is that they operate at near neutral medium, where the dopamine auto-oxidation at the atmosphere may compromise the sensor calibration. Moreover, in the work of Santos et al. [38] dopamine is assayed via its electrochemical oxidation at rather high operating potential, where easily oxidizable organics (e.g., ascorbate) attending a real sample will interfere with the electrode response. On the other hand, Silva and Vieira [39] reported on dopamine quantification by means of square-wave voltammetry under interference-free conditions, applying a large excitation amplitude to achieve a sub-micromolar limit of determination.

Here, we demonstrate how a specifically engineered electrode surface in combination with a laccase possessing high homogeneous activity can be applied in the biosensing of two important NTs—dopamine and L-epinephrine—in concentrations close to the physiological ones (sub-micromolar limits of detection and quantification). The biosensor operates over the range of potentials from 0.2 to −0.1 V (vs. Ag|AgCl, sat. KCl), where the biocatalyzed oxygen reduction reaction takes place, and the two catecholamines play the role of electron-transferring molecules (redox mediators). Furthermore, the studies were performed under acidic conditions that are not only optimal for enzyme activity, but also ensure delayed auto-oxidation of the enzyme substrates when exposed to atmospheric oxygen.

In order to demonstrate the biosensor applicability in pharmaceutical analysis, a real sample—ampoules with dopamine and L-epinephrine injection solutions—have been analyzed under the optimized conditions, and the results manifested a good recovery.

## 2. Materials and Methods

### 2.1. Reagents

Laccase (Lac, E.C. 1.10.3.2) from *Trametes pubescens* was a generous gift from Prof. Dr Roland Ludwig, Department of Food Science and Technology, BOKU–University of Natural Resources and Life Sciences. The lyophilized laccase was dissolved in 0.05 M sodium-citrate buffer, pH = 4, before measurements and had a specific activity of 46 U mg^−1^. One unit is defined as the oxidation of 1.0 μmol of ABTS per min at pH 4 and 30 °C. Cystamine, cysteine (Acros), glutaric aldehyde (Fisher), citric acid monohydrate and sodium citrate (Acros); HAuCl_4_.H_2_O (Acros) were of analytical grade and used as received. Dopamine hydrochloride L-ascorbic acid and L(−)epinephrine (Acros) were used as 10 mM solutions (freshly prepared before each measurement) in citrate buffer, pH = 4.0.

An ampoule with dopamine hydrochloride concentrate for infusion solution (40 mg mL^−1^, WPW Polfa S.A.) and one with adrenaline (L-epinephrine, Sopharma, Bulgaria, concentration 1 mg mL^−1^) were tested as real samples.

The working electrodes were 3 identical glassy carbon discs (d = 3.0 mm, geometric surface area of 0.07 cm^2^, CHI, Austin, TX, USA) in a 6 cm long cylindrical body from chemically resistant PEEK polymer. Buffer solutions (0.05 M) were made of citric acid and sodium citrate dissolved in ultrapure water (B30–Bio, Adrona, Vilnius, Lithuania) with pH = 4.0, adjusted with a pH meter Easy Five (Mettler−Toledo, Columbus, OH, USA). To increase the ionic strength of the buffer solutions, NaClO_4_ (Acros) was added to the buffer to a 0.1 M concentration.

### 2.2. Methods

All electrochemical experiments were performed in a conventional three-electrode cell with working volume of 20 mL, connected to a computer-controlled electrochemical workstation, Autolab PGSTAT 302 N (Metrohm-Autolab, Utrecht, The Netherlands) equipped with NOVA 2.1.5 software. Unmodified or modified glassy carbon electrode was used as working, Ag|AgCl, sat. KCl (Metrohm, Utrecht, The Netherlands)—as a reference—and a platinum foil as an auxiliary electrode. If not otherwise specified, all reported potentials were referred to Ag|AgCl, sat. KCl electrode. Cyclic voltammetry (CV) at a scan rate of 5 mV s^−1^ and differential pulse voltammetry (DPV) at a scan rate of 7 mV s^−1^ with 25 mV amplitude, pulse duration of 50 ms and constant potential chronoamperometry were employed in these studies. Chronoamperometric measurements were performed under constant stirring with a magnetic stirrer (IkaMag RCT, Ika, Staufen, Germany) at a room temperature of 21 ± 1 °C. When necessary, the buffers were purged with either chemically pure argon (99.99%) or air during measurements. All electrochemical measurements were repeated 3 to 5 times with an RSD not exceeding 4.5%.

Electrochemical impedance spectroscopy (EIS) was accomplished in a conventional 3-electrode setup, in 0.1 M KCl containing 5 mM K_4_[Fe(CN)_6_]/K_3_[Fe(CN)_6_] as redox probe over the range of frequencies from 100 kHz to 1 Hz.

The surface morphology of the modified electrode was examined with a scanning electron microscope JEOL JSM-6390 (JEOL, Peabody, MA, USA). No conductive coatings or other treatments were used on the sample prior to SEM observations.

### 2.3. Biosensor Preparation and Validation

Before modification, glassy carbon electrodes were polished with 0.3 and 0.05 μm alumina slurry on a polishing cloth (LECO, Plzeň, Czech Republic), water-rinsed and cleaned by ultrasonication in ultrapure water for 2–3 min.

The working surface of the cleaned and polished glassy carbon electrodes was modified through direct electrodeposition of gold via electroreduction of tetrachloroaurate ion from an electrolyte containing 50 mM HAuCl_4_, dissolved in 0.1 M HCl by cycling between 0 and −0.6 V (vs. Ag|AgCl, sat. KCl) for 1 cycle at a scan rate of 0.1 V s^−1^.

Prior to the enzyme immobilization gold-modified GC electrodes were cleaned electrochemically in 0.5 M H_2_SO_4_ by cyclic voltammetry (CV, scan rate 0.1 V s^−1^) over the potential range from −0.2 to 1.5 V (vs. Ag|AgCl, sat. KCl) for at least 10 cycles, then thoroughly rinsed with ultrapure water.

Laccase was immobilized as described previously [40], through crosslinking of enzyme to cystamine moieties self-assembled on gold-coated glassy carbon using a bi-functional reagent, glutaric aldehyde.

The self-assembly of cystamine was carried out under static conditions by immersing the electrodes in its 10 mg mL^−1^ aqueous solutions. The duration of the sorption process was 2 h. After completing the chemisorption, the loosely bound alkanethiol was removed from the surface by rinsing with ultrapure water. Then, a 3 µL drop of laccase solution (16 mg mL^−1^) was cast on the electrode surface and 1 µL of glutaric aldehyde (45 mM aqueous solution) was mixed with it and allowed to react for at least 30 min at ambient temperature, then another 3 µL drop of laccase solution was cast to which a portion of 1 µL glutaraldehyde was added and allowed to react for another 30 min at ambient temperature. The prepared enzyme electrodes were stored in 0.05 M sodium citrate buffer, pH 4.0, in a refrigerator at 4 °C until use.

The laccase biosensor was validated analyzing solutions with known concentrations: ampoule of dopamine hydrochloride, 40 mg mL^−1^ solution for injections and ampoule containing 1 mg mL^−1^ of L-epinephrine, or adrenaline. The analyses were performed under the optimized conditions and using pre-calibrated electrodes. The samples were spiked in the reaction medium (100 mL for dopamine assay and 20 mL for L-epinephrine assay) in volumes ensuring that the expected concentration will fall within the linear part of the calibration graph. Concentrations were determined on the basis of the calibration plots in molarities and converted to mg mL^−1^ so as to determine the recovery rates.

All the measurements were implemented at least in triplicate to ensure biosensor reproducibility.

## 3. Results and Discussion

### 3.1. Characterization of Modified Electrode

Electrochemical impedance spectroscopy is known as a non-destructive alternating current technique capable of uncovering specific features of the studied systems. Complex plane plots of the impedance for an unmodified glassy carbon electrode and for the same electrode covered with electrodeposited gold are depicted in Figure 1A. Nyquist plot for the bare glassy carbon electrode consists of a semi-circle region characteristic for the charge transfer resistance over the high frequencies range, followed by a linear region tilted at ca. 45°, typical for Warburg impedance, which implies diffusional control over the redox process. The EIS spectrum of gold-modified glassy carbon electrode represents an arch with curvature over the high-frequencies region and near linear trend over the middle- and low-frequencies interval with the angle between the X-axis and the linear part of the spectrum larger than 45 deg. Simulations of the experimental data provide an equivalent Randles circuit (Figure 1B) containing a constant phase element (CPE) connected in parallel to the charge transfer resistance (R_p_). The calculated value for the R_p_ is much smaller than the one for the bare glassy carbon, implying that the electron transfer is facilitated on the gold-modified surface. The CPEs are often treated in the specialized literature as a typical characteristic of imperfect capacitors and suggests considerable surface roughness due to enhanced porosity or nanoparticles—modified electrode surfaces [41].

Scanning electron microscopy studies confirmed that, electrodeposited on glassy carbon, gold forms complex, rosette-like structures with dimensions between 50 and 130 nm. As can be seen from the SEM image (Figure 1C), the modifying gold phase is unevenly distributed, forming cavities sized from several tents to hundreds of nanometers. Visible appearance of the gold layer looks like a lusterless velvet cover with color ranging from brick to light brown. Cyclic voltammograms of the gold-modified electrodes showed a typical pattern of polycrystalline gold with a peak appearing on the reverse scan at ca. 0.9 V (vs. Ag|AgCl, sat. KCl) indicating desorption of surface-bound oxygen. The surface area of gold-covered electrodes was determined to be 0.312 ± 0.009 cm^2^ from calculating the peak area by assuming an amount of 400 µC cm^–2^ electricity to desorb the chemisorbed oxygen [32]. The estimated electrochemically accessible surface area is approximately 12-fold larger than the surface area of the smooth gold electrode with the same diameter.

### 3.2. Voltammetric Behavior of Immobilized Laccase in Aerated and Deaerated Buffer

Laccase immobilization on electrode surfaces is of key importance when aiming at a lasting operational stability of resulting bioelectrodes. To immobilize laccase on the Au electrodeposits, a positively charged at the working pH, self-assembled monolayer, SAM, of 2-aminoethane thiol was created (resulting from the chemisorption of cystamine on gold. This electrode preparation is further denoted in the text as cystamine-modified electrode). Laccase was further confined on the electrode surface by binding it to the thiol layer using the bi-functional agent, glutaric aldehyde.

The electrochemical behavior of the immobilized enzyme was then studied by means of cyclic voltammetry (CV). Comparison of the CVs of the gold-modified electrode with immobilized laccase recorded in the absence and the presence of dioxygen (Figure 2) reveals laccase-catalyzed electrochemical reduction of dissolved oxygen. On the CV recorded with a low scan rate in deaerated buffer (Figure 2, solid line) two broad oxidative peaks appear: at ca. 0.2 V and ca. 0.4 V, whereas on the reverse scan the corresponding reductive peaks were less visible and appeared at potentials of ca. 0.1 V and 0.350 V, respectively. The two pairs of redox peaks are not present on the CV recorded in aerated buffer solution (Figure 2, dash line), however, a clearly expressed reductive wave starts at ca. 0.1 V, which is due to the electrochemical reduction of dissolved oxygen catalyzed by the immobilized enzyme. As already mentioned, laccase is one of the few oxidoreductases capable of exchanging electrons with underlying electrode surfaces without the need for additional electron shuttles (mediators), with the efficiency of the electrical communication controlled by both enzyme orientation and the distance between its active site and the electrode surface. It is hypothesized that the positively charged electrode surface electrostatically attracted the negatively charged laccase active site this way orienting the enzyme in a conformation favorable for electron exchange with the underlying electrode surface, and the voltammetric studies revealed its ability for working in DET mode (direct electron exchange between the enzyme active site and the electrode).

Differential pulse voltammogram (DPV) of the same laccase electrode is depicted in Figure 3 (Figure 3A, curve 1). For comparison, an identically prepared laccase electrode with a SAM of cysteine to which laccase is similarly attached (Figure 3A, curve 2) was also subjected to differential pulse voltammetry. The DPV of laccase immobilized over a cystamine-modified surface (Figure 3A, curve 1) shows a peak at a potential of 0.18 ± 0.02 V that can be assigned to the redox transformation of the T1 copper site, responsible for binding benzenediols, as these observations are in conformity with the literature data [3,42]. In support of this hypothesis is the finding that the DPV of the electrode does not change in the absence of oxygen (deaerated buffer). For the laccase immobilized on the cysteine monolayer (Figure 3A, curve 2), the peak flattens and turns into a shoulder, suggesting a much less efficient electron exchange between the immobilized enzyme active site and the underlying electrode surface. Most probably, cysteine forms zwitterions at the working pH due to the fully dissociated carboxylic terminal group (pK_1_ = 1.71) and protonated vicinal amino group bearing a positive charge [43], thus the cysteine-modified gold forms a layer with both negative and positive charges. The comparison of the DPVs implies the surface charges affect the orientation at which the laccase enzyme lies on the electrode upon its immobilization, and most probably in the case of laccase attached to 2-aminoethane thiol SAM its orientation is favorable for electron exchange with the underlying electrode surface, whereas when attached to a cysteine monolayer the enzyme might experience some steric hindrances when exchanging electrons with the electrode surface, as represented schematically in Figure 3B. The values of the open circuit potentials for the two electrode preparations were found to be similar and around 0.3 V (vs. Ag|AgCl, sat. KCl) at the operating temperature and pH.

### 3.3. Laccase Electrode in the Presence of Dopamine and L-Epinephrine

The appearance of a clearly expressed peak at the DPVs of laccase immobilized on the cystamine-functionalized gold layer, motivated us to further explore the voltammetric behavior of laccase in the presence of two structurally similar catecholamines—dopamine and L-epinephrine. The addition of dopamine aliquots to the buffer followed by the record of the resulting DPV (Figure 4A) caused a notable increase in the peak height even for dopamine concentrations as low as 2 µM. Upon the increase of dopamine concentration, the peak sharpened and shifted to the negative direction. A linear dependence between the peak height and dopamine concentration was observed over the range from 2 up to 40 µM (Figure 4A, Inset), which deviates from linearity at higher concentrations. Under the same conditions, the addition of L-epinephrine (Figure 4B) did not substantially affect the shape or position of voltammetric maxima and the relationship between the peak height and analyte concentration was not consistent.

Further studies of the laccase electrode by cyclic voltammetry (CV), recorded in the presence of each catecholamine (Figure 5) showed additional dissimilarities between the two catecholamines. The presence of both laccase substrates, dioxygen and either catecholamine, caused a reductive current to flow at potentials less positive than 0.4 V (i.e., at much lower overpotentials than the oxygen reduction catalysed by only laccase), thus indicating the ability of these substances to facilitate the electron transfer between the immobilized enzyme and the electrode surface—a phenomenon known as mediated bioelectrocatalysis. Schematically, the mechanism of catecholamine-mediated bioelectrochemical oxygen reduction can be presented as follows:$$\begin{array}{c} E_{red}+O_{2}+4e^{-}+4H^{+}\;\to \;E_{ox}+2H_{2}O\;\\ E_{ox}+M_{red}\;\to \;E_{red}+M_{ox}\\ M_{ox}+e^{-}\overset{electrode}{\to }\;M_{red}, \end{array}$$
where *E_red_* and *E_ox_* stands for the reduced/oxidized form of enzyme and *M_red_* and *M_ox_* the reduced and oxidized forms of catecholamine. The proposed mechanism was confirmed by comparative studies—in the absence of oxygen, no electroreduction of either catecholamine was observed.

The efficiency of the mediated electron transfer (MET process) was found to depend on substrate structure. Immobilized laccase is capable of reducing oxygen in the presence of dopamine at potentials more negative than 0.4 V rather efficiently (Figure 5—red, solid) and reaches a limiting current over the region from 0.3 to 0 V (backward scan) followed by a second reduction wave overlaping the one of enzymatic oxygen reduction in DET mode, however, with higher current intensity. In the presence of L-epinephrine, the efficiency of the MET process over the range 0.4–0 V is far lower (Figure 5—red, dashed line) with the starting potential for the mediated oxygen reduction below 0.3 V (i.e., at ca. 0.1 V less positive potential than with dopamine), and reaching a plateau region below 0 V.

Based on these findings a detailed study on the oxygen reduction mediated by either dopamine or L-epinephrine by means of constant-potential amperometry was performed over the potential region starting from 0.2 down to −0.1 V (vs. Ag|AgCl, sat. KCl). The potential region was selected so that its start point was the potential at which the DPV peak appears, and the end point is the potential at which the reductive current in the presence of L-epinephrine reaches a plateau. Over the whole studied potential region current changes stepwise upon addition of catecholamine stock solution (Figure 6). The time to reach a steady-state current value did not exceed 30 s. As the applied potential shifted to the negative direction, the electrode response to catecholamines increased, reaching the highest electrode sensitivities at potential of 0 V for dopamine and −0.1 V for L-epinephrine. At an applied potential of 0.2 V the biosensor response to L-epinephrine was found to be about twice as low as the one for dopamine, whereas at −0.1 V its sensitivity was 76% of that for dopamine. Such a steep increase in the sensitivity might be a cumulative result from two parallel electrode reactions—mediated oxygen reduction and electroreduction of enzymatically generated L-epinephrine semi-quinone. Electrode sensitivities as a function of the applied potential are presented in Figure 7. It is obvious that the biosensor sensitivity towards dopamine is practically the same at applied potentials of 0 V and −0.1 V and then goes down with increasing operating potential. The same parameter for L-epinephrine gradually decreases with increasing the operating potential over the studied range. As it can be concluded from the discussed findings, the laccase can react with both catecholamines, but possesses different affinity towards them, and therefore their simultaneous determination is not plausible with a single biosensor. However, by using a differential approach, e.g., two identical biosensors poised at different potentials, seems to provide a realistic prospective for discriminating between them.

The electrochemical Michaelis–Menten plots (Figure 8) were drawn on the basis of chronoamperometric records obtained upon additions of catecholamine solutions at a constant potential of –0.1 V (vs. Ag|AgCl, sat. KCl), as the highest electrode sensitivity was achieved at this operating potential. The hyperbolic shape of both dependencies of the current density on substrate concentration suggests that the process is controlled by Michaelis-type enzyme kinetics. For both enzyme substrates, initially the current density raises linearly with catecholamine concentrations up to ca. 120 µM for dopamine and up to ca. 200 µM for L-epinephrine with constant sensitivities of 0.18 µA L mol^−1^ cm^−2^ (dopamine) and 0.12 µA L mol^−1^ cm^−2^ (L-epinephrine). This initial linear trend corresponds to the difusion-controlled region typical for the low substrate concentrations. At higher catecholamine concentrations, the dependence deviates from linearity (i.e., the reaction is controlled by the very enzyme kinetics) and the enzyme reaches saturation at dopamine concentrations exceeding 300 µM, whereas for L-epinephrine the saturation concentration was found to be a bit higher (above 350 µM). The apparent kinetic constants *V_max_* and *K_M_^app^* were calculated from non-linear regression of the experimental data obtained with the two enzyme substrates, and are presented in Table 1. The apparent Michaelis constants differ for the two catecholamines. Its value for dopamine is close to the one determined spectrophotometrically for the native enzyme in solution with the same substrate (*K_M_^app^* = 0.0918 mM as reported in [44]) and the plausible reason for the observed finding is that the immobilized enzyme possesses enough conformational mobility, but the diffusion of the substrate through the enzyme layer most probably enlarges it. 

The apparent Michaelis’ constant for L-epinephrine determined under the same conditions was found to be more than twice as large, thus pointing to a weaker enzyme–substrate interaction. At concentrations exceeding 400 mM a slight decrease of the current density is observed—possibly indicating enzyme inhibition by L-epinephrine.

Operational parameters of the laccase biosensor when either dopamine or L-epinephrine were used as analytes are presented in Table 2. Data analysis suggests greater laccase affinity towards dopamine, which is manifested not only by the kinetic constants, but also by the 1.45-times higher biosensor sensitivity and 1.30-fold larger maximal current density than the for L-epinephrine. The linear dynamic ranges for both analytes span over 0.1 mM, which makes the biosensor a convenient analytical tool for monitoring catecholamines’ concentrations in pharmaceutical products. The limits of detection were calculated as three times the dispersion (σ) of the blank response divided by the slope of the calibration graph (linear part): LOD = 3 σ/slope. The limits of quantification for both analytes were calculated as LOQ = 10 σ/slope, where σ has the same meaning. Both values lay in the sub-micromolar range, which makes the biosensor potentially applicable in biological liquids, as the dopamine physiological levels (10 ng mL^−1^) fall within this range [32].

The discussed here, the operational parameters demonstrate a very good analytical performance of the developed biosensor in terms of electrode sensitivity and limits of detection. A brief comparison of the LOD values for some recently reported biosensors for dopamine (and L-epinephrine) assay is given in Table 3. Surprisingly, the laccase biosensor under study shows LOD values considerably lower [36] than those achieved using electrochemical techniques with forced hydrodynamics (flow-injection analysis, FIA), and comparable values to those achieved with such a highly sensitive pulsed electrochemical technique as square-wave voltammetry (SWV) [39,45].

Validation of the laccase biosensor was implemented by analyzing pharmaceutical products with known concentrations (Table 4). The determined mean values of 41.2 ± 1.5 mg mL^−1^ for dopamine (40 mg mL^−1^) and 0.948 mg mL^−1^ for L-epinephrine (1 mg mL^−1^) support the biosensor applicability for the determination of these two analytes in pharmaceutical products.

### 3.4. Interference Studies and Stability

Having in mind that gold nanoparticles are an excellent catalyst for ascorbate electrochemical oxidation at near zero potential, the presence of L-ascorbic acid in the assayed sample might potentially interfere with the electrode response to either catecholamine. Hence, the laccase electrode response to L-ascorbate was examined under the same experimental conditions at the operating potentials of 0 V and −0.1 V. It was found that at 0 V, the addition of L-ascorbate causes a reductive current to flow at an increased noise level (Figure 9). If present at equal concentrations with L-ascorbic acid the biosensor response to L-epinephrine will add 1.1%, whereas its response to dopamine will be increased by 0.8%, provided that the concentrations were chosen over the linear region of the calibration plots (i.e., up to 0.12 mM). If operating at −0.1 V, the presence of L-ascorbate will affect the biosensor response to dopamine by 1.4%, whereas the response to L-epinephrine will be overestimated by 2.2%. It has to be pointed out that these operating potentials ensure practically interference-free catecholamine determination, since only the presence of L-ascorbate in the sample can potentially interfere with the electrode response to either catecholamine.

Calibration of the laccase electrodes towards either catecholamine was performed under isothermal conditions, at 20 ± 1 °C. The current density of the laccase biosensor in the presence of either catecholamine was found to be highly sensitive to temperature changes: an increase in the temperature by 6–7 deg. caused a drastic decrease in the electrode response. A plausible explanation is the fast auto-oxidation of catecholamines under ambient conditions and thus the substrate solutions were prepared immediately before measurement and kept in an ice bath during chronoamperometric measurements. Some contribution to the decreased reaction rate from the decreased oxygen solubility in buffers as temperature increases is also probable.

Stability of fabricated biosensor was tested over a 26-hour period (Figure 10) using dopamine as the analyte. It was found that the activity of the freshly prepared laccase biosensor remains intact within the next 3 h (consecutive measurements). The maximum current density was accepted as an activity indicator (biosensor response at saturating analyte concentration divided by electrochemically accessible electrode surface). After a 21 h pause, when the biosensor was kept refrigerated in humid atmosphere, the maximum current density decayed to 1/3 of the initial one. It has to be pointed out that a slight increase of the biosensor response to dopamine can be observed after the first use that is most probably due to the facilitated analyte access to the electrode surface as a result of the swelling of the protein-containing layer.

## 4. Conclusions

A conventional glassy carbon electrode has been modified with porous gold through electrodeposition. The resulting modified electrode has an electrochemically accessible surface area that is ca. 12 times larger than the surface area of a smooth gold electrode with the same diameter.The electrochemical behavior of *Trametes pubescens* laccase immobilized on porous gold-modified electrode has been investigated by cyclic and differential pulse voltammetry in both absence and presence of oxygen. Voltammetric studies revealed laccase susceptibility to direct electron transfer and bioelectrocatalytic oxygen reduction. In the presence of catecholamines dopamine and L-epinephrine studies showed the capability of the immobilized laccase to perform mediated oxygen reduction with efficiency dependent on substrate structure.Amperometric measurements performed at a constant potential of −0.1 V (vs. Ag|AgCl, KCl sat.) upon addition of either dopamine or L-epinephrine showed that the dependency of the current density on catecholamine concentrations follows the mechanism of Michaelis–Menten. The apparent kinetic constants have been found to depend on substrate structure with the values of $K_{M}^{app}=0.116\;\pm 0.015\;\mathrm{mM}$ for dopamine and $K_{M}^{app}=0.245\;\pm 0.031\;\mathrm{mM}$ for L-epinephrine. At the potentials of 0 and −0.1 V the electrochemical reduction of the catecholamines semi-quinones is superimposed over ORR, which results in increased electrode sensitivity. The calculated detection limits were found to be in the sub-micromolar concentration range.The concentrations of dopamine hydrochloride and L-epinephrine in ampules with solution for injections has been determined with the developed biosensor. The analytical recovery of the determination has been found to be within 99% and 106% for dopamine and between 89% and 105% for L-epinephrine.Interference studies have shown that at an operating potential of 0 V the presence of L-ascorbate will affect the laccase electrode response to L-epinephrine by 1.1% and to dopamine by 0.8%, whereas at an operating potential of −0.1 V the interference from L-ascorbate will contribute a 2.2% increase in the response to L-epinephrine, and 1.4% to dopamine.Stability tests have shown that after 21 h storage in the refrigerator, the biosensor response to dopamine decays to 1/3 of the initial one.

The discussed experimental findings demonstrate the plausibility of the chosen approach for laccase immobilization for the development of electrochemical biosensors for catecholamines capable of quantifying dopamine or L-epinephrine at close to physiological concentrations.

## Figures and Tables

**Figure 1 biosensors-12-00719-f001:**
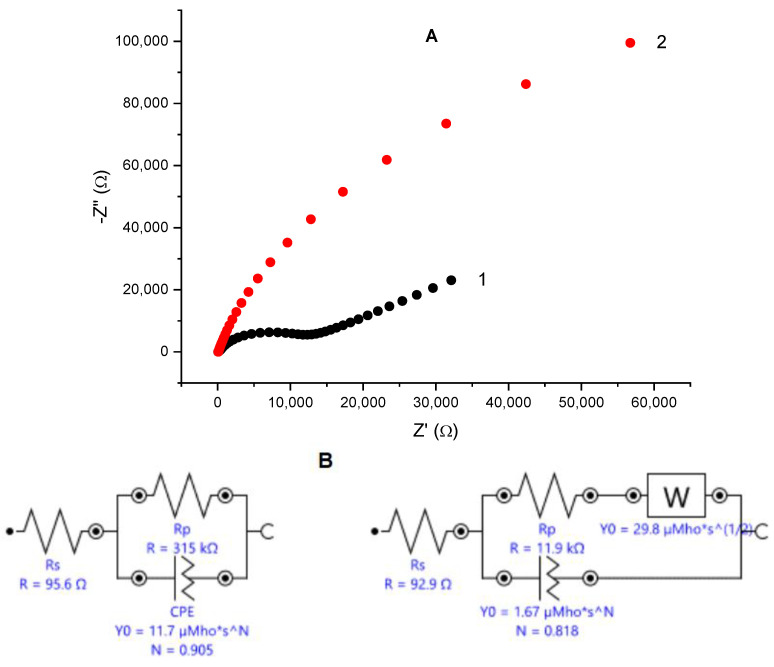
(**A**) Nyquist plot—dependence of the imaginary vs. the real part of the impedance for bare glassy carbon electrode (black squares) and the same electrode after electrodeposition of nanoporous gold (red circles); reference electrode: Ag|AgCl, KCl sat.; electrolyte: 0.1 M KCl containing 5 mM K_4_[Fe(CN)_6_]/K_3_[Fe(CN)_6_] as redox probe; frequency range from 100 kHz to 1 Hz. (**B**) Equivalent Randles circuits for the bare (left) and gold-modified (right) glassy carbon electrode. (**C**) SEM image of electrodeposited gold on glassy carbon surface; magnification ×50,000.

**Figure 2 biosensors-12-00719-f002:**
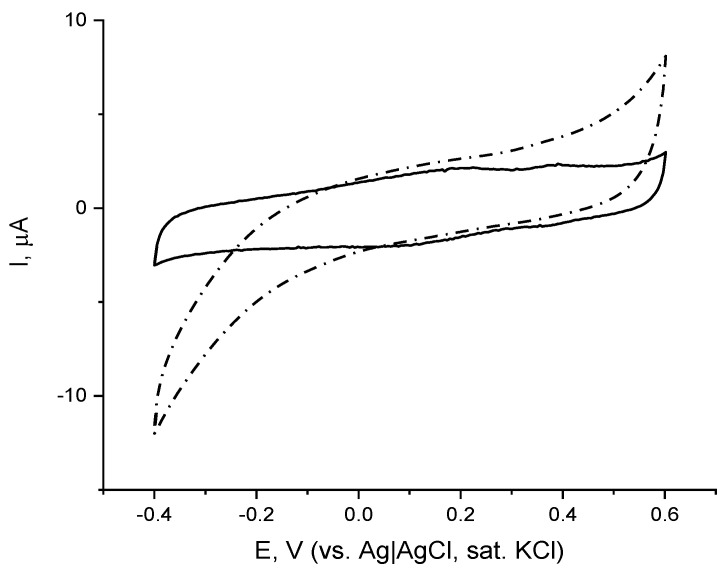
Cyclic voltammograms (second scan) of laccase immobilized on chemisorbed monolayer of cystamine in background electrolyte in aerated (dash) and de-aerated with Ar medium (solid), 0.05 M citrate buffer containing 0.1 M NaClO_4_, pH = 4.0; scan rate, v = 5 mV/s, reference electrode Ag|AgCl, sat. KCl.

**Figure 3 biosensors-12-00719-f003:**
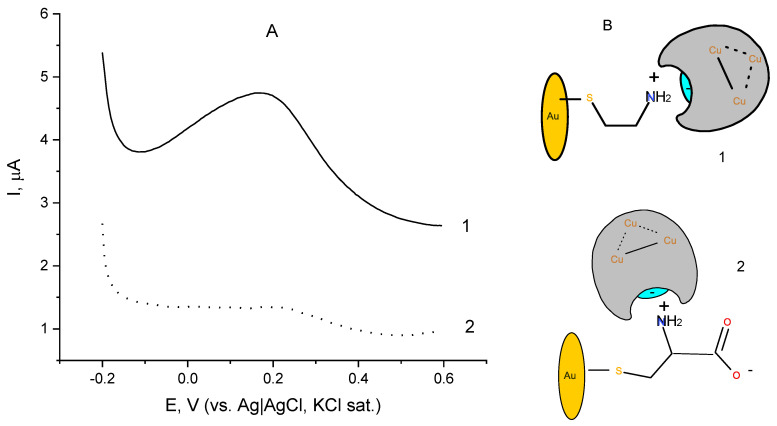
(**A**) Differential pulse voltammograms of laccase immobilized on chemisorbed monolayer of (1) cystamine and (2) cysteine; background electrolyte: 0.05 M citrate buffer, pH 4 containing 0.1 M NaClO_4_; (**B**) Schematic of laccase orientation upon immobilization over a self-assembled monolayer of (1) cystamine and (2) cysteine residues.

**Figure 4 biosensors-12-00719-f004:**
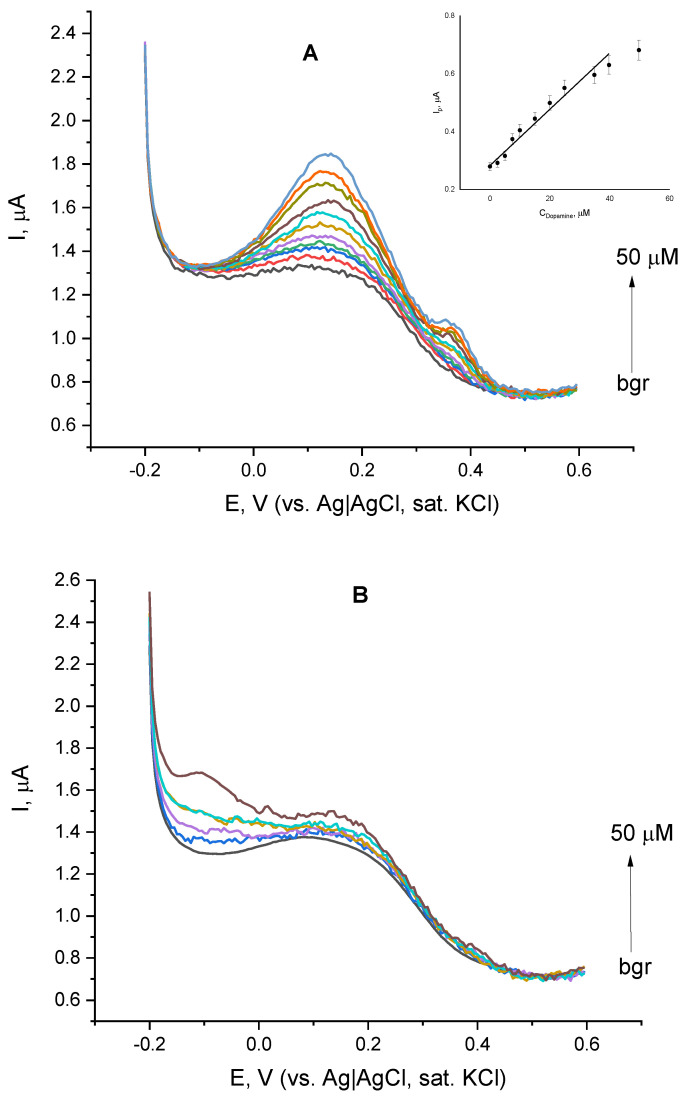
Differential pulse voltammograms of laccase immobilized on cystamine-modified gold in the presence of dopamine (**A**) and L-epinephrine (**B**) with concentrations ranging from 0 to 50 µM; background electrolyte—constantly aerated 0.05 M citrate buffer containing 0.1 M NaClO_4_, pH = 4.0. Inset of (**A**) dependence of the DPV peak height on dopamine concentration.

**Figure 5 biosensors-12-00719-f005:**
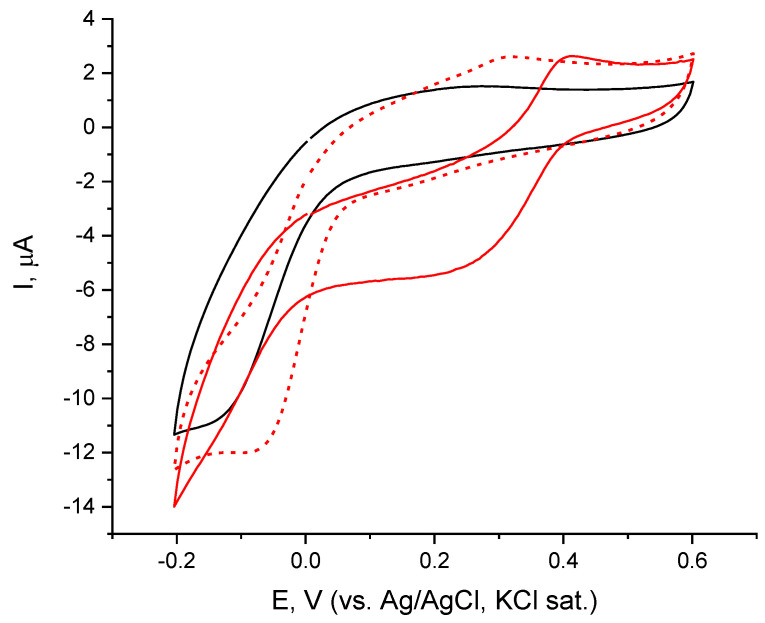
Cyclic voltammograms (second scan) of laccase electrode immobilized on chemisorbed monolayer of cystamine in background electrolyte (black, solid) and in dopamine (red, solid) and L-epinephrine (red, dash) present; concentration of catecholamines, C = 49.8 µM.

**Figure 6 biosensors-12-00719-f006:**
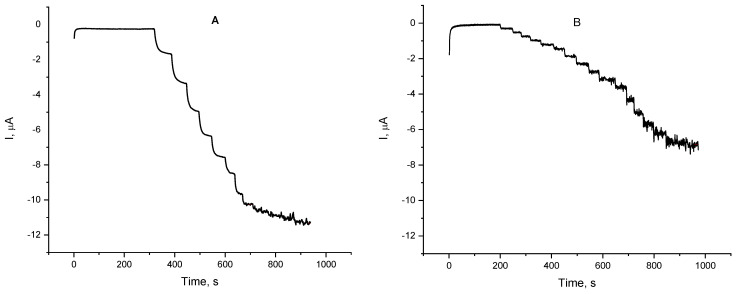
Chronoamperometric records of the laccase biosensor response to the addition of aliquots of dopamine (**A**) and L-epinephrine (**B**) stock solution (10 mM in citrate buffer pH = 4.0); electrolyte—0.05 M citrate buffer with 0.1 M NaClO_4_, pH = 4.0 working potential −0.1 V (vs. Ag|AgCl, sat. KCl); 20 ± 1 °C.

**Figure 7 biosensors-12-00719-f007:**
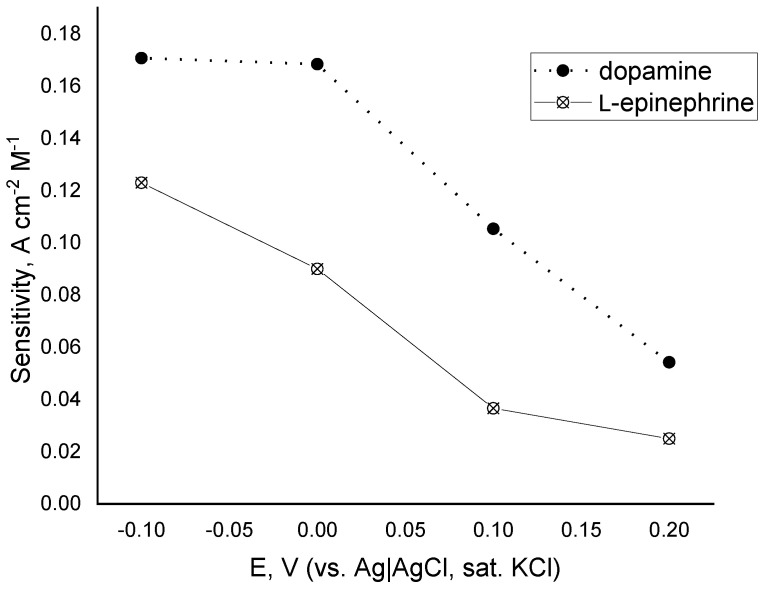
Sensitivity of the determination of dopamine (closed circles) and L-epinephrine (open circles) vs. operating potential over the potential region from −0.1 to 0.2 V (vs. Ag|AgCl, sat. KCl); electrolyte—0.05 M citrate buffer with 0.1 M NaClO_4_, pH = 4.0; 20 ± 1 °C.

**Figure 8 biosensors-12-00719-f008:**
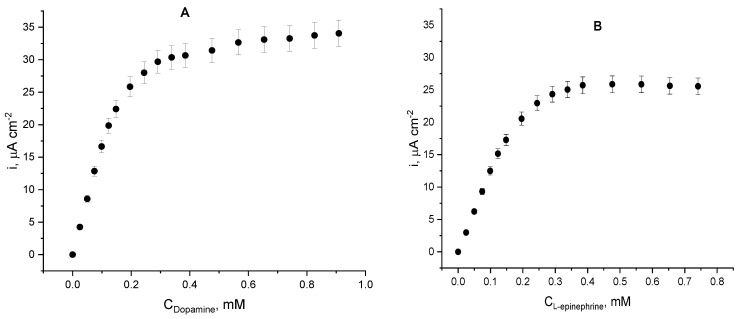
Dependence of the current density on catecholamine concentrations (electrochemical Michaelis–Menten plot); working electrodes: laccase immobilized on cystamine monolayer in the presence of dopamine (**A**) and L-epinephrine (**B**); citrate buffer, pH = 4.0; working potential −0.1 V (vs. Ag|AgCl, sat. KCl); 20 ± 1 °C.

**Figure 9 biosensors-12-00719-f009:**
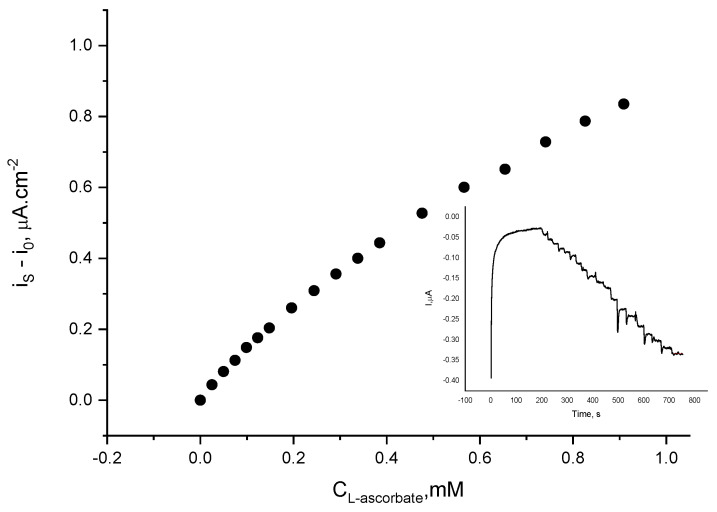
Current density variation of laccase biosensor upon addition of L-ascorbic acid aliquots at 0 V (vs. Ag|AgCl, sat. KCl); electrolyte: 0.05 M citrate buffer with 0.1 M NaClO_4_, pH = 4.0; temperature 21 ± 1 °C. Inset: Authentic record.

**Figure 10 biosensors-12-00719-f010:**
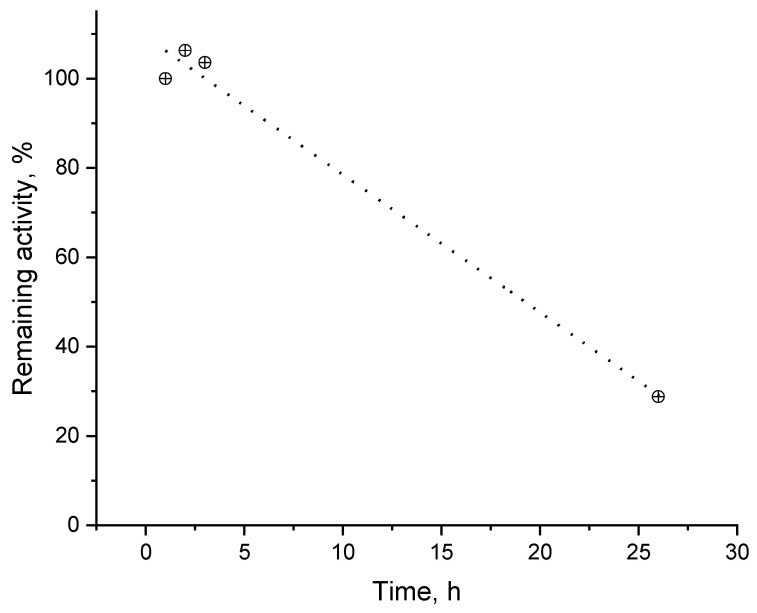
Remaining catalytic activity as a function of the time passed since biosensor fabrication. The data were normalized to the first measurement at E = −0.1 V in citrate buffer, pH = 4.0: substrate—dopamine.

**Table 1 biosensors-12-00719-t001:** Kinetic constants of the immobilized laccase calculated from non-linear least-square regression by fitting the data to a hyperbolic function based on chronoamperometric plots recorded at an operating potential of −0.1 V.

*Kinetic Constants*	*Dopamine*	*L-Epinephrine*
*V_max_^app^*, (A cm^−2^)	(4.08 ± 0.16) × 10^−5^	(3.36 ± 0.17) × 10^−5^
*K_M_ ^app^* (mM)	0.116 ± 0.015	0.245 ± 0.031
Coefficient of determination, R^2^	0.999	0.998

**Table 2 biosensors-12-00719-t002:** Operational parameters of laccase biosensor in dopamine and L-epinephrine present as substrates; operating potential −0.1 V (vs. Ag|AgCl, sat. KCl); temperature: 20 ± 1 °C.

*Operational Parameters*	*Dopamine*	*L-Epinephrine*
Maximum current density i_max_ (Acm^−2^)	(33.76 ± 1.2) × 10^−6^	(25.80 ± 1.3) × 10^−6^
Sensitivity A L mol^−1^ cm^−2^	0.178 ± 0.005	0.123 ± 0.002
Linear dynamic range, mM	0.12	0.19
Limit of detection LOD, M	3.74 × 10^−8^	5.41 × 10^−8^
Limit of quantification, LOQ, M	1.25 × 10^−7^	1.80 × 10^−7^

**Table 3 biosensors-12-00719-t003:** Comparison of detection limits for some recently reported laccase-based biosensors for dopamine.

*Biosensor Type*	*Method of Analysis*	*Analyte*	*LOD,* *μM*	*Reference*
Lac-mesoporous silica biosensor	Amperometry (FIA)	Dopamine L-Epinephrine	5.46 15.5	[36]
Lac-Glu-AuNPs/CPE	Amperometry (E = 0.3 V)	Dopamine	0.06	[38]
AuNP-PAH-LAC/CPE	SWV	Dopamine	0.26	[39]
Lac-HNT-ImS3–14/CPE	SWV	Dopamine	0.252	[45]
Lac-GA-NH_2_C_2_H_4_S-AuNS/GC	Amperometry (E = −0.1 V)	Dopamine L-Epinephrine	0.037 0.054	This work

**Table 4 biosensors-12-00719-t004:** Biosensor validation with real samples—injection solutions with known concentration of dopamine (40 mg mL^−1^) and L-epinephrine (1 mg mL^−1^).

*Analyte*	*Spiked Volume,* *μL*	*Concentration Determined, mg mL^−1^*	*Recovery, %*
**Dopamine**	10 20 30	39.6 42.3 41.5	99 106 104
**L-Epinephrine**	20 40 60	0.89 1.046 0.909	89 105 91

## Data Availability

Not applicable.

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
