# Peer review of "Biosensing Dopamine and L-Epinephrine with Laccase (*Trametes pubescens*) Immobilized on a Gold Modified Electrode"

_biosensors, 2022, doi:10.3390/bios12090719_

Round 1

Reviewer 1 Report

This work reports the development of an electrochemical biosensor based on laccase-catalyzed immobilized with cystamine on gold-coated glassy carbon electrode for the determination of dopamine and L-epinephrine. The biosensor is not easy to build and is not stable. Stability tests have shown that after 21 hours of storage in the refrigerator, the biosensor response to dopamine decays to 1/3 of the initial one. The method has been validated for dopamine only. Assays for L-epinephrine are lacking. There are some clarifications to be made. Therefore, I recommend that a major review be done so that it can be considered for publication in Biosensors.

General comments:

1. The novelty of the work needs to be clarified. Similar sensors for dopamine using laccase and gold nanoparticles have already been published. See at: Analyst, 141 (2016) 216-224 and C – Journal of Carbon Research, 8 (2022) 40.

2. The biosensor manufacturing process is quite labor intensive. How many measurements can be performed with just one biosensor? that is, without the need to prepare a new one. In addition, the biosensor is not stable (21 hours).

3. The authors, like many others, confuse the terms "detection" and "determination". Detection is qualitative by nature, while determination always is quantitative. Qualitative analysis is the detection of the presence of ions or compounds in an unknown sample, for example. The term "determination" refers to quantitative analysis to obtain data on the amount of analyte by weight or by concentration of an element or a compound in a sample. Therefore, most of the words “detection" in the manuscript should be replaced by the term "determination" (or "quantitation" or "assay") if quantitative assays are involved.

4. The correct name of R2 is coefficient of determination. Check the information. Read more in Critical Reviews in Analytical Chemistry, 36 (2006) 41–59.

Specific comments:

1. Materials and methods:

a. Line 124. The DPV has three parameters: scan speed, pulse amplitude and pulse duration time. Provide pulse duration time.

b. Line 137. Replace "electrode preparation" with "biosensor preparation"

c. Lines 146-149. Was gold oxide formation evaluated after pre-cleaning the GCE modified with gold nanoporous in sulfuric acid?

2. Results and discussion:

a. Section 3.1. The study of impedance with the different steps of modification of the electrode is incomplete. After the electrodeposition of the gold nanoporous, the electrode is still immersed in cystemine, then laccase solution is dripped, then glutaraldehyde solution is dripped, and then more laccase and glutaraldehyde. So, provide Nyquist plots of all modification steps.

b. Line 189. Where are the cyclic voltammograms?

c. Line 193. The electroactive area must be estimated via the Randles-Sevcik equation or by EIS.

d. A pH study should be performed for dopamine and L-epinephrine. For evaluation of different pH values, the Britton-Robinson buffer solution can be used.

e. It is not possible to perform a simultaneous determination of dopamine and L-epinephrine. There is an overlap of the peaks. Comment about this fact must be inserted.

f. Lines 388-391. The data of the determination of dopamine in the injection samples must be presented in a table. Provide the value found in the real sample, the value added (spiked), the found value after spiked, and the recovery values (%).

g. Provide a table comparing the different detection limits for dopamine and L-epinephrine and their different laccase-based biosensors. Some studies to be added: Analyst, 141 (2016) 216-224; C – Journal of Carbon Research, 8 (2022) 40 and Biochemical Engineering Journal, 186 (2022) 108565.

h. The biosensor was not applied to real samples containing L-epinephrine. Thus, accuracy data were not provided.

i. Provides intra-day and inter-day repeatability studies for dopamine and L-epinephrine. Thus, the accuracy of the data can be discussed.

j. Lines 464-465. What are the physiological concentrations found of dopamine and L-epinephrine? Are the LOQ values sufficient for this purpose?

Author Response

Authors appreciate the referee’s time and efforts spent in reviewing our work. From the reviewer’s comments it looks like their expertise is mainly in electroanalytical science (statistics and validation), however enzyme-based biosensors probably are not in their main focus of their research. Authors have taken into account all the reviewer’s comments and critical notes and elaborated the corresponding changes in the revised MS. A rebuttal against two of the required changes is also provided. The corrections in the MS made during the revision process are highlighted. More details were provided in the attached pdf-file.

Reviewer 2 Report

Pimpilova et al. demonstrated the application of gold modified electrode in detection of dopamine and L-epinephrine. Author report on the development and optimization of an electrochemical biosensor based on laccase from Trametes pubescens immobilized on glassy carbon electrode modified with nanoporous gold. Here we demonstrate how a specifically engineered electrode surface in combination with a laccase possessing high homogeneous activity can be applied in biosensing of two important NTs dopamine and L-epinephrine in concentrations close to the physiological ones (sub-micromolar limits of detection and quantification). Some points need to be addressed:

1.     Author should provide the response of sensor in presence of other molecules to prove the selectivity of developed sensor.

2.     Please provide the comparative table in order to compare the efficiency of present sensor with already reported sensors such as IEEE Transactions on Biomedical Engineering 67 (6), 1542-1547, 2019.

3.     Please mention about the detection time required in each case.

4.     In figure 1 there is some problem in labelling.

5.      Please elaborate the mechanistic part.

Author Response

Authors appreciate the reviewer’s time and efforts during the Peer-review process. All their helpful comments and suggestions were taken into account and elaborated during the revision of the MS. The changes in the MS made are highlighted.

Reviewer 3 Report

This manuscript entitled “Biosensing dopamine and L-epinephrine with laccase (Trametes pubescens) immobilized on gold modified electrode” by Mariya Pimpilova et al. reports a dopamine and L-epinephrine sensing method by using electrochemical sensor that uses nanogold-coated carbon working electrode with laccase modification. The nanogold deposition was verified by SEM imaging and EIS measuring. The laccase immobilization on nanogold surface with different crosslinker was investigated and validated by CV and DPV. For dopamine and L-epinephrine biosensing, constant-potential amperometry were implemented to study the calibration curve and the linearly sensing range was built. Overall, the theory and sensing mechanism are well described. The manuscript should be accepted after adding some experiments and answering some questions.  

It seems that the dopamine and the L-epinephrine both can react with laccase. The authors investigated the interference and discussed in the manuscript. However, for a real biosensor, the authors may need to provide a sensing protocol, and use the proposed protocol to determine samples containing with well-known concentrations of dopamine and the L-epinephrine.

Please the authors may comment the electrochemical biosensing with nanogold electrode, made by different ways in the section of introduction. Nanogold surface can be made e.g.  

(1)   Peptide-based electrochemical sensor with nanogold enhancement for detecting rheumatoid arthritis” Talanta, 236, 122886 1 January, 2022. (DOI: 10.1016/j.talanta.2021.122886)

Minor problem

Fig 4 label of (A) and (B) are missing.  

Author Response

Authors are thankful to this reviewer for the attention given to their manuscript and for helpful comments. All the Reviewer’s suggestions were taken into account. The changes in the MS made during the revision process are highlighted. Detailed answers of the items pointed out from the Reviewer are available in the attached pdf-file.

Round 2

Reviewer 1 Report

The manuscript was improved and the authors clarified the doubts. Therefore, I recommend that the article be accepted for publication in Biosensors.

Reviewer 3 Report

It can be published in current form